# One-Token Rollout: Guiding Supervised Fine-Tuning of LLMs with Policy Gradient

## Abstract

Supervised fine-tuning (SFT) is the predominant method for adapting large language models (LLMs), yet it often struggles with generalization compared to reinforcement learning (RL). In this work, we posit that this performance disparity stems not just from the loss function, but from a more fundamental difference: SFT learns from a fixed, pre-collected dataset, whereas RL utilizes on-policy data sampled from the current policy. Building on this hypothesis, we introduce one-token rollout (OTR), a novel fine-tuning algorithm that guides SFT with the policy gradient method. OTR reframes the autoregressive learning process by treating each token generation as a single-step reinforcement learning trajectory. At each step, it performs a Monte Carlo "rollout" by sampling multiple candidate tokens from the current policy's distribution. The ground-truth token from the supervised data is then used to provide a reward signal to these samples. Guided by policy gradient, our algorithm repurposes static, off-policy supervised data into a dynamic, on-policy signal at the token level, capturing the generalization benefits of on-policy learning while bypassing the costly overhead of full sentence generation. Through extensive experiments on a diverse suite of challenging benchmarks spanning mathematical reasoning, code generation, and general domain reasoning, we demonstrate that OTR consistently outperforms standard SFT. Our findings establish OTR as a powerful and practical alternative for fine-tuning LLMs and provide compelling evidence that the on-policy nature of data is a critical driver of generalization, offering a promising new direction for fine-tuning LLMs.

## 1 Introduction

Supervised fine-tuning (SFT) has become a cornerstone for adapting large language models (LLMs) to downstream tasks (Ouyang et al., 2022; Chung et al., 2022; Zhang et al., 2025). However, a growing body of evidence suggests that while SFT excels at mimicking expert demonstrations, it often struggles with generalization compared to methods based on reinforcement learning (RL) (Chu et al., 2025a; Huan et al., 2025; Shenfeld et al., 2025). Recent research Chu et al. (2025a) has proposed the view that "SFT memorizes, while RL generalizes". This limitation is particularly concerning as SFT can disrupt the well-formed distributions learned during pre-training, leading to a degradation of general capabilities—a phenomenon sometimes referred to as catastrophic forgetting (Kumar et al., 2022; Huan et al., 2025; Shenfeld et al., 2025).

This generalization gap motivates a deeper investigation into the fundamental differences between SFT and RL, with the goal of enhancing the generalization of SFT by borrowing principles from RL. Recent advancements in RL have demonstrated that even simplified methods, such as GPG (Chu et al., 2025b), which directly optimize an objective structurally similar to a weighted SFT loss, can achieve performance comparable to more complex algorithms like PPO (Schulman et al., 2017) or GRPO (Shao et al., 2024). This suggests that the performance disparity between SFT and RL may not solely stem from the loss function, but also from a more fundamental difference: the nature of the data used for updates. SFT typically relies on a static, pre-collected set of expert demonstrations, which is known as off-policy data, whereas RL methods utilize on-policy data sampled iteratively from the current policy.

As RL becomes an increasingly popular paradigm for fine-tuning LLMs, the critical role of on-policy data has garnered significant attention (Tajwar et al., 2024; Ren & Sutherland, 2024; Shenfeld et al.,

2025). Tajwar et al. (2024) has shown that on-policy sampling is crucial for RL to discover optimal policies, especially when the target behavior lies in low-probability regions of the initial model. It provides a more stable and effective learning signal by ensuring that policy updates are made in regions the model can already reach, thereby preventing drastic and potentially harmful shifts in the output distribution (Ren & Sutherland, 2024; Shenfeld et al., 2025). This suggests that the on-policy nature of RL is a key factor contributing to its superior generalization and ability to preserve pre-trained knowledge.

Inspired by these insights, we propose one-token rollout (OTR) algorithm, a novel fine-tuning method that aims to enhance the generalization of SFT from a data-centric perspective. OTR guides the fine-tuning process with the policy gradient method, treating each token-generation step as an individual, on-policy learning event. By performing a Monte Carlo "rollout" at each token position which samples candidate tokens from the current policy and using the ground-truth token as a reward signal, OTR transforms the off-policy supervised data into a token-level on-policy signal.

OTR enhances generalization by narrowing the data-side gap between SFT and RL, while its design as a token-level method bypasses the costly generation of complete, sentence-level on-policy training data. Our extensive experiments demonstrate that this on-policy simulation consistently improves the generalization of fine-tuned models across a wide array of challenging mathematical, coding, and general reasoning benchmarks. These results not only validate the efficacy of OTR as a powerful alternative for fine-tuning LLMs but also provide strong evidence for the critical role that on-policy data plays in the generalization performance of fine-tuned language models.

Our contributions can be summarized as follows:

- We introduce One-Token Rollout, a novel fine-tuning algorithm that guides SFT with the policy gradient method. By treating each token generation as a single-step reinforcement learning task, OTR improves model generalization without incurring the high computational cost of full sentence generation.

- We provide a new data-centric perspective on the SFT-RL generalization gap, positing that the on-policy nature of training data is a critical factor. The success of our token-level on-policy simulation serves as strong evidence for this viewpoint.

- We conduct extensive experiments on a wide array of challenging benchmarks across mathematical, coding, and general reasoning domains. Our results empirically demonstrate that OTR consistently outperforms SFT, validating its efficacy as a powerful and practical alternative for fine-tuning LLMs.

## 2 PRELIMINARIES

### 2.1 SUPERVISED FINE-TUNING

The standard approach for adapting pre-trained LLMs to specific downstream tasks is Supervised Fine-Tuning. Given a dataset of prompt-response pairs, where the response $X$ is a sequence of tokens $\{x_1, x_2, \ldots, x_T\}$, SFT aims to maximize the conditional probability of the ground-truth sequence. This is typically achieved by minimizing the negative log-likelihood loss, autoregressively training the model $\pi_\theta$ to predict the next token $x_t$ given the preceding context $x_{1:t-1}$:

$$\mathcal{L}_{\text{SFT}}(\theta) = -\frac{1}{T} \sum_{t=1}^{T} \log \pi_\theta(x_t | x_{1:t-1}). \tag{1}$$

### 2.2 POLICY GRADIENT

Policy gradient represents a class of reinforcement learning algorithms that directly optimize a parameterized policy, $\pi_\theta$. In this framework, the text generation process is modeled step-by-step. At each timestep $t$, the state $s_t$ is the sequence of previously generated tokens $x_{1:t-1}$, and the action $a_t$ is the next token selected by the policy from the vocabulary.

The core objective is to adjust the policy's parameters, $\theta$, to maximize the expected total reward. This objective function, $J(\theta)$, is defined as the expected cumulative reward:

$$J(\theta) = \mathbb{E}_{\tau \sim \pi_\theta} \left[ \sum_{t=1}^{T} r(s_t, a_t) \right], \tag{2}$$

where $r(s_t, a_t)$ is the scalar reward received after taking action $a_t$ in state $s_t$, and $\tau$ is the entire sequence of states and actions $(s_1, a_1, s_2, a_2, \dots)$, known as a trajectory.

The policy is improved by ascending the gradient of this objective, $\nabla_\theta J(\theta)$. The foundational policy gradient theorem provides a way for the gradient computation:

$$\nabla_\theta J(\theta) = \mathbb{E}_{\tau \sim \pi_\theta} \left[ \left( \sum_{t=1}^{T} \nabla_\theta \log \pi_\theta(a_t \mid s_t) \right) \left( \sum_{t=1}^{T} r(s_t, a_t) \right) \right], \tag{3}$$

where $\nabla_\theta \log \pi_\theta(a_t \mid s_t)$ indicates the direction in the parameter space that would be used to update the policy $\pi_\theta$. This direction is then weighted by the sum of all rewards in the trajectory, effectively reinforcing action sequences that lead to higher overall rewards.

## 3 METHODOLOGY

We introduce the One-Token Rollout algorithm, a novel fine-tuning method that adapts the principles of Policy Gradient to the token level. OTR reframes the standard fine-tuning process by treating each individual token generation step as a complete, single-step trajectory. This conceptual shift allows us to simplify the general policy gradient framework into a highly efficient, token-level reinforcement learning algorithm, where the supervised training data is repurposed to provide a reward signal.

### 3.1 FROM POLICY GRADIENT TO ONE-TOKEN ROLLOUT

Our starting point is the foundational policy gradient theorem introduced in the Section 2. The core innovation of OTR is to consider the generation of a single token from a state $s_t$ to an action $a_t$ as a complete trajectory of length one. In this micro-trajectory, the summations over timesteps present in Equation (3) collapse, as there is only a single state-action pair. Consequently, the summations of $\nabla_\theta \log \pi_\theta(a_t \mid s_t)$ and $r(s_t, a_t)$ over $T$ tokens in the original formula both reduce to terms for an individual token, and sampling a full trajectory $\tau$ simplifies to sampling a single action $a_t$ from the policy $\pi_\theta(\cdot | s_t)$. The policy gradient for this single step thus simplifies dramatically to:

$$\nabla_\theta J(\theta) = \mathbb{E}_{a_t \sim \pi_\theta(\cdot | s_t)} \left[ \nabla_\theta \log \pi_\theta(a_t \mid s_t) \cdot r(s_t, a_t) \right]. \tag{4}$$

To implement this, we approximate the expectation $\mathbb{E}[\cdot]$ using Monte Carlo estimation. At each timestep $t$ of the original sequence, we perform a "rollout" by sampling multiple candidate actions from the current policy. This transforms the optimization problem into a practical, sample-based loss function.

### 3.2 TOKEN-LEVEL ROLLOUT AND ON-POLICY REWARD

To facilitate the rollout, we first define a stochastic sampling policy and a reward mechanism.

**Stochastic Policy for Exploration**. For a given state $s_t$, the LLMs first compute a vector of raw, unnormalized scores for every token in the vocabulary $V$. These scores are known as logits. Let $l_a$ denote the logit corresponding to a specific action $a$. The model's base policy, $\pi_\theta$, is typically derived by applying the softmax function directly to these logits.

To encourage exploration during the rollout, we create a new sampling policy, $\pi'_\theta$, by introducing a temperature parameter $\kappa$. The sampling policy is defined as:

$$\pi'_\theta(a | s_t) = \text{softmax} \left( \frac{l_a}{\kappa} \right). \tag{5}$$

Consistent with its common use during model inference, the temperature adjusts the shape of the final probability distribution. We utilize a temperature $\kappa > 1$ to flatten the distribution, which increases the likelihood of sampling less probable tokens and thereby enhances exploration.

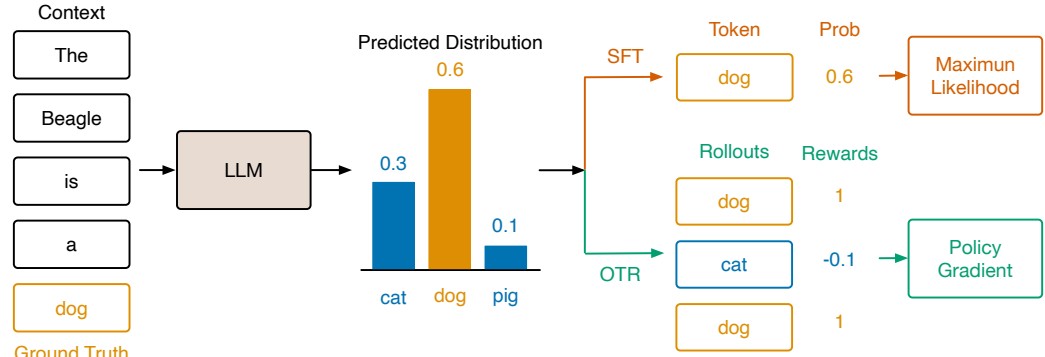

Figure 1: An illustration of the computational divergence between SFT and OTR.

**Rollout and Reward Definition**. At each timestep $t$, we draw a set of $K$ candidate actions from our exploration policy:

$$\mathcal{A}'_t = \{a'_{t,j}\}_{j=1}^K, \quad \text{where each } a'_{t,j} \sim \pi'_\theta(\cdot|s_t). \tag{6}$$

Crucially, we use the ground-truth token $x_t$ from the supervised dataset to construct an immediate reward signal. Each sampled action $a'_{t,j}$ is evaluated against $x_t$ using the following reward function:

$$R(a'_{t,j}, x_t) = \begin{cases} 1 & \text{if } a'_{t,j} = x_t, \\ \beta & \text{if } a'_{t,j} \neq x_t. \end{cases} \tag{7}$$

Here, $\beta$ is a hyperparameter where $\beta < 1$. A reward of 1 is given for "rediscovering" the ground-truth token, while a lesser reward $\beta$ is given for all other tokens. We finally set $\beta = -0.1$ for our main experiments based on the ablation study detailed in Section 4.4.

This design elegantly converts the traditionally off-policy supervised data into an on-policy learning signal at the token level. The actions $\mathcal{A}'_t$ we evaluate are sampled directly from the current policy $\pi'_\theta$, and the fixed ground-truth token $x_t$ is used simply to assign a real-time reward to these on-policy actions. This avoids the complexities of importance sampling or other off-policy correction techniques typically required in sentence-level RL.

### 3.3 THE OTR OBJECTIVE FUNCTION

Based on the token-level rollout and policy gradient in Equation (4), the loss at timestep $t$ is the Monte Carlo approximation of the negative policy gradient objective, averaged over the $K$ samples:

$$\mathcal{L}^t_{\text{OTR}}(\theta) = -\frac{1}{K} \sum_{j=1}^K \left[ \text{sg}\left(R(a'_{t,j}, x_t)\right) \cdot \log \pi_\theta(a'_{t,j}|s_t) \right], \tag{8}$$

where $\pi_\theta$ is the model's original, non-temperature-scaled policy, and $\text{sg}(\cdot)$ is the stop-gradient operator. Given our defined reward function, we can decompose this loss. Let $N_{gt} = \sum_{j=1}^K \mathbb{I}(a'_{t,j} = x_t)$ be the count of times the ground-truth token was sampled. The loss function simplifies to its final form:

$$\mathcal{L}^t_{\text{OTR}}(\theta) = -\frac{1}{K} \left[ N_{gt} \log \pi_\theta(x_t|s_t) + \beta \sum_{j \text{ s.t. } a'_{t,j} \neq x_t} \log \pi_\theta(a'_{t,j}|s_t) \right]. \tag{9}$$

This per-timestep objective has an intuitive interpretation. The first term is a SFT-like loss for the ground-truth token, but it is dynamically weighted by its sampling frequency $N_{gt}$. If the ground-truth is never sampled, its loss contribution is zero. The second term acts as a regularizer, weighted by $\beta$, which penalizes the model for assigning high probability to the incorrect tokens it sampled.

The total loss for an entire sequence of length $T$ is the average of these per-timestep losses:

$$\mathcal{L}_{\text{OTR}}(\theta) = \frac{1}{T} \sum_{t=1}^T \mathcal{L}^t_{\text{OTR}}(\theta). \tag{10}$$

This objective allows OTR to focus its optimization effort, reinforcing correct predictions that are already within the model's reach while gently suppressing plausible alternatives, creating a more nuanced and effective learning signal than SFT alone. To visually summarize the computational divergence of the OTR update from the strandard SFT, we provide a detailed illustration in Figure 1.

### 3.4 COMPARISON WITH DYNAMIC FINE-TUNING

Our work is related to the concurrent dynamic fine-tuning (DFT) method (Wu et al., 2025c), which also seeks to improve the generalization of SFT from a reinforcement learning perspective. DFT's motivation stems from the insight that the standard SFT gradient contains an implicit, problematic inverse-probability weighting ($1/\pi_\theta$) that leads to optimization instability. To address this, DFT proposes to "rectify" the reward by reweighting the loss for the ground-truth token $x_t$ with its own model probability $\pi_\theta(x_t|s_t)$. The resulting per-timestep DFT loss is:

$$\mathcal{L}_{\text{DFT}}^t(\theta) = -\text{sg}(\pi_\theta(x_t|s_t)) \log \pi_\theta(x_t|s_t). \tag{11}$$

The OTR framework can be seen as a generalization of DFT. This relationship becomes clear when we consider the special case of our OTR objective where the hyperparameter $\beta = 0$. In this scenario, the second term in Equation (9), which penalizes incorrect samples, vanishes. Then the OTR loss can be formulated as:

$$\mathcal{L}_{\text{OTR}}^t(\theta)|_{\beta=0} = -\frac{N_{gt}}{K} \log \pi_\theta(x_t|s_t), \tag{12}$$

where $N_{gt}/K$ represents the empirical frequency of sampling the ground-truth token during the rollout. This frequency is a direct Monte Carlo approximation of the ground-truth token's probability, i.e., $\frac{N_{gt}}{K} \approx \pi_\theta(x_t|s_t)$. Thus, when $\beta = 0$, the OTR objective is functionally equivalent to the DFT objective, as both methods effectively weight the loss of the ground-truth token by its estimated probability.

However, when $\beta \neq 0$, OTR extends beyond DFT's formulation. In addition to reinforcing the "rediscovered" ground-truth token, OTR's objective incorporates a crucial second term: a regularization penalty applied to the incorrect tokens sampled during the rollout. This allows OTR to not only learn from the positive signal of the ground-truth but also to actively discourage the model from assigning high probability to plausible but incorrect alternatives. Therefore, OTR provides a more comprehensive learning signal by leveraging information from both successful and unsuccessful samples within the model's own distribution.

## 4 EXPERIMENTS

### 4.1 EXPERIMENT SETTINGS

**Dataset and Models**. We conduct experiments on the OpenR1-Math-220k dataset (OpenR1 Team, 2025), which consists of 220,000 mathematical problems with detailed reasoning traces. These traces are generated by the DeepSeek R1 model (DeepSeek-AI et al., 2025) for problems originating from the NuminaMath-1.5 dataset (LI et al., 2024). To efficiently manage computational resources while ensuring data quality, we randomly sample a subset of 5,000 instances for our training set. All selected instances have reasoning traces with lengths under 8192 tokens, and their lengths are approximately uniformly distributed across different intervals. We utilize a suite of powerful and contemporary open-source LLMs as base models. Specifically, we conduct our experiments on the following models: Qwen2.5-3B (Qwen Team, 2024), Qwen2.5-7B (Qwen Team, 2024), Qwen3-4B-Base (Qwen Team, 2025), and Qwen3-8B-Base (Qwen Team, 2025).

**Training Details**. Our implementation is built upon the Verl framework, and to ensure a fair comparison, both our proposed OTR algorithm and the SFT baseline are trained using identical settings. We employ the AdamW optimizer with a learning rate of $5 \times 10^{-6}$. The learning rate follows a cosine decay schedule, which includes a warm-up ratio of 0.03 and decays to $1 \times 10^{-6}$. For the training configuration, we use a batch size of 64 and a maximum sequence length of 10240 tokens. All models are trained for a total of 2 epochs.

Table 1: Main results on in-domain mathematical reasoning benchmarks. For each model, the best result between SFT and OTR is in **bold**. The †symbol indicates performance degradation compared to the base model.

| Model | Method | GSM8K | MATH | Olympiad | Minerva | AIME24 | AIME25 | AMC23 | Average |
|-------|--------|-------|------|----------|---------|--------|--------|-------|---------|
| Qwen2.5-3B | Base | 77.90 | 42.64 | 25.20 | 23.20 | 3.30 | 0.00 | 40.00 | 30.32 |
| | SFT | 82.05 | 62.50 | 26.23 | 24.90 | 7.30 | 1.65 | 37.03† | 34.52 |
| | OTR | **82.93** | **63.95** | **27.05** | **25.00** | **7.71** | **2.91** | **40.78** | **35.76** |
| Qwen2.5-7B | Base | 85.36 | 49.80 | 36.40 | 28.30 | 6.70 | 3.30 | 42.50 | 36.05 |
| | SFT | 88.18 | 67.75 | 31.53† | 32.53 | **8.54** | 5.00 | 43.75 | 39.61 |
| | OTR | **89.77** | **70.45** | **35.33**† | **33.45** | 8.33 | **6.87** | **44.38** | **41.23** |
| Qwen3-4B | Base | 86.90 | 54.10 | 38.20 | 29.80 | 3.30 | 6.70 | 55.00 | 39.14 |
| | SFT | 74.13† | 63.95 | 32.10† | 29.60† | 10.21 | 6.24† | 42.66† | 36.98† |
| | OTR | **91.98** | **75.30** | **40.63** | **36.68** | **10.22** | **11.67** | **52.81**† | **45.61** |
| Qwen3-8B | Base | 90.40 | 60.80 | 40.90 | 34.20 | 13.30 | 16.70 | 62.50 | 45.54 |
| | SFT | 83.77† | 77.40 | 41.70 | 37.70 | **15.20** | **15.63**† | 55.16† | 46.65 |
| | OTR | **91.63** | **79.45** | **42.43** | **39.35** | 14.80 | 14.17† | **59.53**† | **48.77** |

Table 2: Out-of-domain performance on code generation and general reasoning benchmarks. For each model, the best result between SFT and OTR is in **bold**. The †symbol indicates performance degradation compared to the base model.

| Model | Method | Code | | | General Reasoning | | | |
| | | HumanEval+ | MBPP+ | Avg | BBEH | SuperGPQA | MMLU-Pro | Average |
|-------|--------|------------|-------|-----|------|-----------|----------|---------|
| Qwen2.5-3B | Base | 35.40 | 50.30 | 42.85 | 6.00 | 19.28 | 33.90 | 19.73 |
| | SFT | 57.60 | 48.20† | 52.90 | 7.23 | 18.67† | 36.15 | 20.68 |
| | OTR | **59.30** | **49.90**† | **54.60** | **7.88** | **19.22**† | **36.20** | **21.10** |
| Qwen2.5-7B | Base | 48.80 | 64.00 | 56.40 | 6.88 | 23.93 | 42.31 | 24.37 |
| | SFT | 68.50 | 58.10† | 63.30 | 10.44 | 26.25 | **51.24** | 29.31 |
| | OTR | **69.00** | **59.20**† | **64.10** | **11.28** | **26.32** | 51.22 | **29.61** |
| Qwen3-4B | Base | 56.70 | 62.40 | 59.55 | 8.19 | 28.56 | 53.35 | 30.03 |
| | SFT | 70.20 | 60.90† | 65.55 | 9.27 | 28.11† | 53.86 | 30.41 |
| | OTR | **74.00** | **62.90** | **68.45** | **9.71** | **29.03** | **55.96** | **31.57** |
| Qwen3-8B | Base | 61.60 | 63.50 | 62.55 | 9.91 | 32.53 | 59.57 | 34.00 |
| | SFT | 76.00 | 65.40 | 70.70 | **10.40** | 29.03† | 53.82† | 31.08† |
| | OTR | **77.70** | **66.50** | **72.10** | 10.02 | **30.49**† | **56.87**† | **32.46**† |

## 4.2 EVALUATION

Our evaluation is designed to accurately reflect the impact of the SFT and OTR fine-tuning algorithms on the base models' capabilities. To this end, we utilize a suite of challenging benchmarks spanning mathematical, code, and general reasoning domains to test the generalization of the algorithms, and we employ distinct evaluation settings for the base and fine-tuned models. For all evaluations, the maximum generation length is set to 8192 tokens.

**Benchmarks and Metrics**. Our evaluation covers a suite of challenging benchmarks across three domains. For **mathematical reasoning**, our evaluation includes Minerva Math (Lewkowycz et al., 2022), MATH-500 (Hendrycks et al., 2021), GSM8K (Cobbe et al., 2021), OlympiadBench (He et al., 2024), AMC 2023, AIME 2024, and AIME 2025. For the highly challenging AMC 2023, AIME 2024, and AIME 2025 benchmarks, we report **mean@16** accuracy, while for the remaining math benchmarks, we report **mean@4** accuracy. For **code generation**, we use HumanEval Plus (Liu et al., 2023) and MBPP Plus (Liu et al., 2023), with performance measured by the **pass@1** metric. Finally, for **general domain reasoning**, we evaluate on MMLU-Pro (Wang et al., 2024), SuperG-PQA (Du et al., 2025a), and BBEH (Kazemi et al., 2025) using **Exact Match (EM)** accuracy.

**Base Model Evaluation**. To align with standard evaluation practices for base models, we use a natural prompt template for testing. Specifically, we employ a 5-shot setting for the MATH-500 and

Table 3: Ablation study on the hyperparameter $\beta$ for in-domain mathematical reasoning.

| Model | Method | | GSM8K | MATH | Olympiad | Minerva | AIME24 | AIME25 | AMC23 | Average |
|---|---|---|---|---|---|---|---|---|---|---|
| Qwen2.5-3B | SFT | | 82.05 | 62.50 | 26.23 | 24.90 | 7.30 | 1.65 | 37.03 | 34.52 |
| | OTR $\beta =$ | -1.00 | 83.10 | 63.05 | 26.05 | **25.75** | 6.88 | **2.91** | 37.66 | 35.06 |
| | | -0.10 | 82.93 | **63.95** | 27.05 | 25.00 | 7.71 | **2.91** | 40.78 | **35.76** |
| | | 0.00 | 83.65 | 63.10 | **27.48** | 22.35 | **8.34** | 2.69 | 39.53 | 35.31 |
| | | 0.01 | **83.75** | 63.70 | 27.30 | 24.65 | 5.43 | 2.69 | 36.72 | 34.89 |
| Qwen3-4B | SFT | | 74.13 | 63.95 | 32.10 | 29.60 | 10.21 | 6.24 | 42.66 | 36.98 |
| | OTR $\beta =$ | -1.00 | **92.15** | **77.75** | 40.60 | 35.68 | **12.09** | **13.33** | **53.75** | **46.48** |
| | | -0.10 | 91.98 | 75.30 | 40.63 | 36.68 | 10.22 | 11.67 | 52.81 | 45.61 |
| | | 0.00 | 91.03 | 76.30 | **40.75** | 36.88 | 10.20 | 10.63 | 53.28 | 45.58 |
| | | 0.01 | 90.15 | 76.30 | 39.35 | **36.95** | 9.79 | 9.79 | 51.41 | 44.82 |

GSM8K benchmarks and use a greedy sampling strategy with a temperature of 0 for decoding for all benchmarks.

**Fine-tuned Model Evaluation**. For the chat models fine-tuned with SFT and OTR, we use their respective chat templates and a 0-shot setting across all benchmarks. The decoding strategy is stochastic sampling with a temperature of 0.7 and a top-p of 0.8.

### 4.3 RESULTS

We present the main experimental results in Table 1 for in-domain generalization and Table 2 for out-of-domain (OOD) generalization. For all OTR experiments presented in this section, we set the key hyperparameters for our algorithm: the temperature parameter $\kappa = 1.3$, the number of rollout candidates $K = 256$, and the reward hyperparameter $\beta = -0.1$. The value for $\beta$ was determined to yield the best overall performance based on our ablation studies detailed in Section 4.4.

**In-Domain Generalization**. As shown in Table 1, OTR consistently demonstrates superior performance over SFT on mathematical reasoning tasks. Across all four model families, OTR achieves a higher average score. This highlights OTR's effectiveness in enhancing the specialized capabilities of the models within their training domain.

Furthermore, OTR shows greater generalization by mitigating the catastrophic forgetting often observed during fine-tuning. The number of instances where performance degrades below the base model (marked by the [†]symbol) is significantly lower for OTR (4 instances) compared to SFT (10 instances). Even in cases where both methods underperform, OTR's performance drop is considerably milder. For example, on the AMC23 benchmark with Qwen3-8B, SFT's score drops by 7.34 points relative to the base model, whereas OTR's score drops by only 2.97 points. This suggests that OTR's on-policy signal helps preserve the valuable knowledge learned during pre-training.

**Out-of-Domain Generalization**. The advantages of OTR extend to out-of-domain tasks, as detailed in Table 2. On both code generation and general reasoning benchmarks, OTR consistently surpasses SFT in average performance across all models. This trend demonstrates that OTR effectively leverages its on-policy signal to achieve broader, more generalized capabilities that are not confined to its training domain.

From the perspective of knowledge preservation, OTR again proves to be a more generalizable algorithm. SFT underperforms its base model in 7 OOD instances, particularly showing vulnerability on SuperGPQA and MMLU-Pro with larger models. In contrast, OTR underperforms in 5 instances and shows consistent improvements on general reasoning for the Qwen3-4B model where SFT struggles. This demonstrates that OTR provides a more reliable fine-tuning approach that not only enhances target skills but also better maintains the model's general intelligence, leading to superior overall generalization. A supplementary experiment is detailed in Appendix A.

### 4.4 ABLATION STUDY

To investigate the impact of the reward hyperparameter $\beta$, we conduct a comprehensive ablation study. We select four values for analysis, ranging from negative to positive: -1.0, -0.1, 0, and 0.01. The performance across in-domain and out-of-domain benchmarks is presented in Table 3

Table 4: Ablation study on the hyperparameter $\beta$ for out-of-domain generalization.

| Model | Method | | | Code | | | General Tasks | | |
|---|---|---|---|---|---|---|---|---|---|
| | | | HumanEval+ | MBPP+ | Avg | BBEH | SuperGPQA | MMLU-Pro | Average |
| | SFT | | 57.60 | 48.20 | 52.90 | 7.23 | 18.67 | 36.15 | 20.68 |
| Qwen2.5-3B | OTR $\beta =$ | -1.00 | 57.90 | 49.70 | 53.80 | 7.28 | 19.07 | 36.26 | 20.87 |
| | | -0.10 | 59.30 | **49.90** | 54.60 | 7.88 | 19.22 | 36.20 | 21.10 |
| | | 0.00 | 58.90 | 49.20 | 54.05 | **8.38** | **19.39** | 36.16 | **21.31** |
| | | 0.01 | **60.50** | **49.90** | **55.20** | 7.59 | 19.08 | **36.49** | 21.05 |
| | SFT | | 70.20 | 60.90 | 65.55 | 9.27 | 28.11 | 53.86 | 30.41 |
| Qwen3-4B | OTR $\beta =$ | -1.00 | **74.20** | 61.10 | 67.65 | **9.91** | 28.65 | **56.51** | **31.69** |
| | | -0.10 | 74.00 | **62.90** | **68.45** | 9.71 | **29.03** | 55.96 | 31.57 |
| | | 0.00 | 73.70 | 61.90 | 67.80 | 9.54 | 28.24 | 53.93 | 30.57 |
| | | 0.01 | 73.20 | 62.20 | 67.70 | 8.38 | 26.11 | 50.37 | 28.29 |

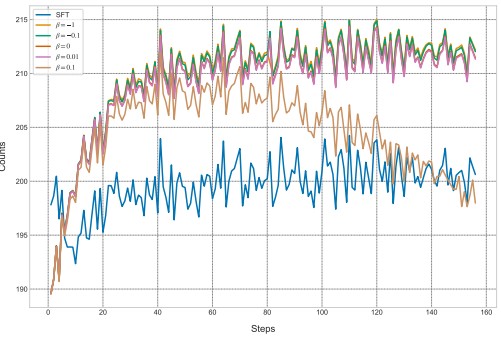

(a) Effect of $\beta$ on GT token counts.

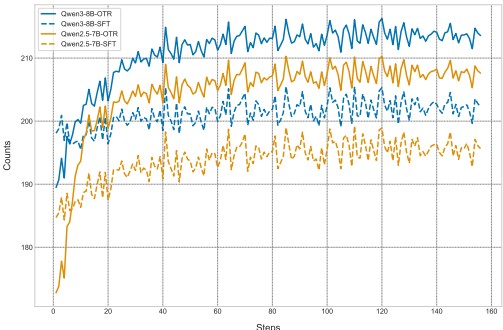

(b) GT token counts for larger models.

Figure 2: Analysis of the number of GT tokens sampled during training. (a) Compares OTR with different $\beta$ values against SFT on Qwen3-4B. (b) Compares OTR and SFT on larger models.

and Table 4, respectively. To provide insight into the training process for our subsequent analysis, we also track a key diagnostic metric: the number of ground-truth (GT) tokens sampled during the token-level rollout. For a direct comparison, we also record this metric for SFT. It is important to note that this measurement is for analysis only and does not alter the standard SFT algorithm.

**Effect of $\beta$ on Training Stability**. Our first key observation relates to training stability, as depicted in Figure 2(a). While most OTR variants show a stable increase in GT token counts, the setting with $\beta = 0.1$ exhibits clear training instability. Its GT count initially rises but then collapses in the later stages. We hypothesize that assigning a positive reward to incorrectly sampled tokens, especially a relatively high one, can mislead the optimization process. This may cause the model to increase the probabilities of all rolled-out tokens indiscriminately, ultimately leading to a degradation of the learned distribution. This observed instability motivates us to limit our search space, leading to our selection of $\beta$ values {-1.0, -0.1, 0, 0.01}, which primarily explores the non-positive range.

**Impact on Performance and Optimal $\beta$ Selection**. From the performance results in Table 3 and Table 4, it is evident that OTR is robustly superior to SFT. Regardless of the specific $\beta$ value, OTR variants consistently outperform the SFT baseline in terms of average scores across nearly all domains and models. Among these variants, the setting of $\beta = -0.1$ demonstrates the most consistent and high-level performance across both in-domain and OOD tasks. Therefore, we select $\beta = -0.1$ as the default value for our main experiments.

**The Importance of Negative Samples**. This study also provides insight into the importance of utilizing negative samples. As analyzed in Section 3.4, OTR with $\beta = 0$ can be viewed as a Monte Carlo approximation of the DFT method. A direct comparison between the $\beta = -0.1$ and $\beta = 0$ rows in our tables reveals that the former almost universally outperforms the latter. This result provides empirical evidence that incorporating an explicit penalty for negatively sampled tokens is a crucial component of OTR's success, contributing to a more effective learning signal than what is offered by SFT-like formulations.

**Analysis of Learning Dynamics**. Finally, we analyze the source of OTR's general superiority over SFT by examining the GT token counts at convergence in Figure 2. Across different models, scales, and architectures (as shown in both Figure 2(a) and Figure 2(b)), OTR-trained models consistently converge to a higher number of sampled GT tokens than SFT-trained models. A higher GT count indicates that the model's learned policy assigns a higher probability to the ground-truth sequences, which suggests a lower perplexity on the training data. We infer from this that OTR enables the model to learn from and utilize the training data more profoundly and efficiently than SFT, potentially unlocking a higher performance ceiling.

## 5 RELATED WORK

**Reinforcement Learning for Language Models**. Recently, reinforcement learning has gained significant traction as a powerful paradigm for enhancing the capabilities of large language models (Hu, 2025; DeepSeek-AI et al., 2025; Wu et al., 2025a). The success of state-of-the-art models, which have leveraged RL-based algorithms like GRPO (Shao et al., 2024) to achieve substantial improvements in reasoning and cross-domain generalization, has catalyzed a surge of interest in these methods. The traditional approach to RL fine-tuning, reinforcement learning from human feedback (RLHF) (Ouyang et al., 2022), often relies on complex and computationally intensive algorithms like PPO (Schulman et al., 2017). The inherent instability and implementation complexity of PPO have motivated a recent wave of research focused on simplifying the RLHF pipeline. A prominent line of work, including methods like DPO (Rafailov et al., 2023) and GEPO (Wu et al., 2025b), elegantly reframes the preference learning objective to create a simple loss, eliminating the need for an explicit reward model. In a similar spirit, GPG (Chu et al., 2025b) simplifies the RL objective into a weighted maximum likelihood form, demonstrating that such a simplified approach can match the performance of more complex algorithms.

**Improving Supervised Fine-tuning**. While SFT is the most widely used paradigm for fine-tuning, its limitations, such as catastrophic forgetting and deviation from the pre-trained model's distribution, are well-documented (Kumar et al., 2022; Huan et al., 2025). A major line of research aims to improve SFT by modifying its objective function. A prominent example is proximal SFT (Zhu et al., 2025), which introduces a proximal regularization term to the SFT loss to penalize divergence from the initial model's policy. This approach is analogous to the KL-divergence constraint in PPO and helps stabilize training and preserve pre-trained knowledge. Another significant line of work seeks to enhance SFT by reformulating it through the lens of reinforcement learning, often by establishing a mathematical connection between their objectives. For instance, some studies reframe RLHF as a reward-weighted form of SFT (Du et al., 2025b), while others view SFT as an RL method with an implicit reward function (Wang et al., 2025; Qin & Springenberg, 2025). Concurrent to our work, DFT (Wu et al., 2025c) identifies an implicit inverse-probability weighting in the SFT gradient and addresses the resulting instability by re-weighting the loss for the ground-truth token with its own model probability. Although these works build a theoretical bridge between SFT and RL, they primarily focus on re-weighting the loss for the static, ground-truth expert data. In contrast, our work offers a distinct, data-centric solution. OTR moves beyond loss modification and instead transforms the training data itself into a dynamic, on-policy signal by actively sampling from the model's current policy.

## 6 CONCLUSION

In this work, we investigated the generalization weakness of SFT compared to RL, positing that the disparity stems from the fundamental difference between SFT's static, off-policy data and RL's dynamic, on-policy data. To bridge this gap from a data-centric perspective, we introduced One-Token Rollout, a novel fine-tuning algorithm. By reframing each token generation as a single-step reinforcement learning trajectory, OTR transforms the static supervised dataset into a dynamic, on-policy learning signal, successfully incorporating the advantage of on-policy data into the SFT framework while maintaining its computational efficiency. Our extensive experiments empirically validate this approach, demonstrating that OTR consistently outperforms SFT on a wide array of in-domain and out-of-domain benchmarks. Ultimately, we present OTR as a powerful and practical alternative for fine-tuning LLMs, providing compelling evidence that simulating on-policy interaction is a key direction for developing more generalizable fine-tuned language models.

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

# A  ADDITIONAL EXPERIMENT

To assess the robustness of our method and validate its generalization benefits across different training data and training configurations, we conduct an additional experiment. For this analysis, we adopt the training data and hyperparameter settings from the concurrent work, dynamic fine-tuning (Wu et al., 2025c), which provides a distinct training environment to test the efficacy of OTR. This comparative analysis focuses on the Qwen2.5-3B (Qwen Team, 2024) and Qwen3-4B (Qwen Team, 2025) models, with the detailed setup provided below.

**Dataset**. We train with the NuminaMath CoT dataset (LI et al., 2024), which comprises around 860,000 mathematical problems paired with corresponding solutions. To efficiently manage computational resources, we randomly sample 50,000 instances from this dataset for training.

**Training Details**. Our implementation is built upon the Verl framework. For a fair comparison, both our proposed OTR algorithm and the SFT baseline are trained using identical settings. Specifically, we employ the AdamW optimizer with a peak learning rate of $5 \times 10^{-5}$. The learning rate follows a cosine decay schedule with a warm-up ratio of 0.1. We use a batch size of 256, a maximum input length of 4096 tokens, and train all models for 1 epoch.

As shown in Table 5, even under the training settings adapted from DFT, our OTR method consistently outperforms the standard SFT baseline across the majority of benchmarks. This finding demonstrates the robustness of the OTR algorithm and suggests that its generalization benefits are not confined to a specific set of data and hyperparameters but hold true across different settings.

Table 5: Results of SFT and OTR on in-domain math benchmarks when trained under the DFT experimental settings. For each model, the best result is in **bold**.

| Model | Method | GSM8K | MATH | Olympiad | Minerva | AIME24 | AIME25 | AMC23 | Average |
|-------|--------|-------|------|----------|---------|--------|--------|-------|---------|
| Qwen2.5-3B | SFT | 78.50 | 53.25 | 19.43 | 16.55 | 2.28 | 0.83 | 24.53 | 27.91 |
|  | OTR | **78.70** | **57.10** | **21.53** | **21.50** | **2.70** | **1.86** | **28.75** | **30.31** |
| Qwen3-4B | SFT | **88.75** | 64.80 | 30.60 | **27.30** | 6.25 | 4.38 | 35.78 | 36.84 |
|  | OTR | 88.05 | **68.65** | **33.88** | 25.90 | **9.38** | **6.46** | **42.66** | **39.28** |

# LIMITATIONS

While our experiments demonstrate OTR's consistent advantages across a range of models and benchmarks, this work has several limitations. First, due to computational constraints, our study is conducted on models up to 8 billion parameters and trained on a subset of a mathematics-focused dataset. Consequently, the scalability of OTR to larger-scale models (e.g., 70B+) remains to be validated. Second, our investigation is confined to the text-only modality. The reward mechanism, while effective, is also relatively simple. Future work will aim to address these limitations by scaling OTR to larger models, training on larger datasets, and extending it to broader training domains. We also plan to explore more sophisticated reward functions, investigate the potential of multi-token rollouts, and extend the OTR framework to other modalities, such as vision-language tasks.

# DECLARATION OF LLM USAGE

The usage of LLMs is strictly limited to aid and polish the paper writing.

