# OpenReview forum: "One-Token Rollout: Guiding Supervised Fine-Tuning of LLMs with Policy Gradient"
_ICLR.cc/2026/Conference — ICLR 2026 Conference Withdrawn Submission_

### Official Review · Reviewer_mP3t · 2025-10-18

**Soundness:** 1
**Presentation:** 4
**Contribution:** 2
**Rating:** 2
**Confidence:** 4

**Summary:**

This paper introduces OTR, a training algorithm based on supervised data inspired in RL that treats each token generation as a single-step rollout of the policy. It then uses a reward function that takes the value 1 if the correct token was sampled and a hyperparameter $\beta$ otherwise.

**Strengths:**

S1) The paper is extremely well-written and clear
S2) The proposed algorithm is shown to improve results on a range of models and reasoning datasets.

**Weaknesses:**

The main weakness of this paper is that the claim that OTR is a generalization of DFT is false. This method is a Monte-Carlo approximation of the DFT objective which can be computed without the additional variance. The reason is the following. Take the OTR objective, replacing the empirical estimate of the expectation with the expectation is approximating:
1) $$\frac{1}{T}\sum_{t=1}^T E_{a_t\sim \pi_\theta(\cdot|s_t)} {R(a_t, x_t)\nabla_\theta  \log \pi_\theta(a_t|s_t)}$$

Now, consider (a constant baseline):
2) $$\frac{1}{T}\sum_{t=1}^T E_{a_t\sim \pi_\theta(\cdot|s_t)} {\beta \nabla_\theta  \log \pi_\theta(a_t|s_t)} = 0$$

This is because $E_{a_t\sim \pi_\theta(\cdot|s_t)} {\beta \nabla_\theta  \log \pi_\theta(a_t|s_t)} = E_{a_t\sim \pi_\theta(\cdot|s_t)} {\beta \frac{\nabla_\theta \pi_\theta(a_t|s_t)}{ \pi_\theta(a_t|s_t)}} = \beta \sum_a{\nabla_\theta \pi_\theta(a|s_t)} = \beta \nabla_\theta(1) = 0$

Thus, subtracting 2 from 1, we get:
3) $$\frac{1}{T}\sum_{t=1}^T E_{a_t\sim \pi_\theta(\cdot|s_t)} {(R(a_t, x_t) - \beta) \nabla_\theta  \log \pi_\theta(a_t|s_t)}$$

Now, because now all actions $a_t \neq x_t$ have reward 0, we are left with:
4)$$ = \frac{1}{T}\sum_{t=1}^T (1-\beta) \pi_\theta(x_t|s_t) \nabla_\theta  \log \pi_\theta(x_t|s_t)$$

Discarding the proportionality constant we are left with
5) $$ \propto \frac{1}{T}\sum_{t=1}^T \pi_\theta(x_t|s_t) \nabla_\theta  \log \pi_\theta(x_t|s_t)$$

which is the DFT loss multiplied by an additional factor $\frac{1}{T}$. This factor, however, is biasing the objective, see: Liu, Zichen, et al. "Understanding r1-zero-like training: A critical perspective." arXiv preprint arXiv:2503.20783 (2025).

The above analysis also shows that whatever effect seen by setting different values of $\beta$ should disappear by dividing the learning rate by $1 - \beta$.

Now, I am not saying that because this paper boils down to a high-variance version of DFT it shouldn't be published. DFT is concurrent work and in the current landscape of Machine Learning with new papers pouring out with breakneck speed every week, it is fair to recognize two contributions coming out simultaneously. However, if I haven't made a mistake in my analysis, and please correct me if I did, the paper should be thoroughly restructured in order to acknowledge these points.

**Questions:**

I don't need to ask any further clarifying questions

---

> ### Author Response · Authors · 2025-11-27
>
> We thank **Reviewer mP3t** for their compliments on the clarity of our writing and the recognition of our empirical improvements across various reasoning tasks.
>
> We appreciate the reviewer's attempt to analyze the theoretical properties of our objective. However, we must respectfully point out a **fundamental mathematical error** in the reviewer's derivation, specifically in the assumption that the $\beta$-term constitutes a "constant baseline" that integrates to zero. This misunderstanding leads to an incorrect conclusion regarding the equivalence of OTR and DFT.
>
> We address the weaknesses below.
>
> ___
> > Weaknesses: The main weakness of this paper is that the claim that OTR is a generalization of DFT is false... Take the OTR objective, replacing the empirical estimate of the expectation with the expectation is approximating... Now, consider (a constant baseline): 2) ... This is because ... = 0
>
> - **Mathematical Error 1: Incorrect Summation Domain:** The reviewer's proof relies on the identity $\sum_{a} \nabla \pi_\theta(a|s_t) = \nabla(1) = 0$. However, this identity *only* holds when the summation is performed over the **entire vocabulary** and the coefficient (baseline) is constant for *all* $a$.
>     - In OTR, the reward $\beta$ is assigned **conditionally**, specifically only to **non-ground-truth tokens** ($a_t \neq x_t$).
>     - Therefore, the expectation term for $\beta$ involves a summation over $a \neq x_t$, not the full vocabulary.
>     - Mathematically: $\sum_{a \neq x_t} \nabla \pi_\theta(a|s_t) = \nabla (1 - \pi_\theta(x_t|s_t)) = -\nabla \pi_\theta(x_t|s_t) \neq 0$.
>     - Consequently, the term does **not** vanish to zero. It produces a non-zero gradient that actively penalizes the probability of incorrect tokens, which is a structural difference from DFT.
>
> - **Mathematical Error 2: Ignoring the Sampling Mechanism:** The reviewer's formulation $\frac{1}{T} \sum_{t=1}^{T} \mathbb{E}_{a_t \sim \pi}[\dots]$ implicitly assumes an expectation over the full distribution. This completely overlooks the core algorithmic design of OTR: **Sampling $K$ Candidates**.
>     - OTR calculates the loss based on a sampled subset $\mathcal{A}'$ of size $K$. Even if we consider the expectation over this subset, the sum of probabilities of sampled tokens $\sum_{a \in \mathcal{A}'} \pi_\theta(a|s_t)$ is strictly less than 1.
>     - The reviewer's derivation implies $\nabla \sum_{a \in \mathcal{A}'} \pi_\theta(a) = 0$, which is mathematically incorrect because the subset $\mathcal{A}'$ is a random variable, not the fixed full vocabulary.
>
> - **Conclusion:** The reviewer's derivation fails because it treats a **conditional, subset-based reward** as a **global constant baseline**. By ignoring the piecewise nature of the reward function (1 for GT, $\beta$ for non-GT) and the restricted domain of the summation (sampled candidates excluding GT), the proof collapses.
>
> We respectfully request the reviewer to re-examine the OTR algorithm. The sampling of $K$ candidates is a crucial step that ensures **on-policy exploration**, and the resulting gradient is distinct from DFT. Given that the rejection rating was primarily based on this mathematical misunderstanding, we hope the reviewer will reconsider the score.

---

### Official Review · Reviewer_z5Y3 · 2025-10-25

**Soundness:** 2
**Presentation:** 3
**Contribution:** 1
**Rating:** 2
**Confidence:** 4

**Summary:**

This paper proposes One-Token Rollout (OTR), a novel algorithm for supervised fine-tuning of LLMs that incorporates principles from policy gradient methods in RL. The authors argue that the generalization gap between SFT and RL stems primarily from SFT's use of static, off-policy data, as opposed to RL's on-policy sampling. OTR addresses this by reframing each token prediction as a single-step RL trajectory, performing Monte Carlo rollouts to sample candidate tokens from the model's current distribution, and using the ground-truth token to assign rewards (1 for matches, beta < 1 for mismatches). This transforms SFT data into a token-level on-policy signal, leading to a loss that dynamically weights the ground-truth and penalizes sampled incorrect tokens. Experiments on mathematical reasoning, code generation, and general benchmarks using Qwen-series models show OTR outperforming standard SFT, with better generalization and reduced catastrophic forgetting.

**Strengths:**

- OTR demonstrates consistent improvements over SFT baselines across diverse benchmarks, including in-domain math tasks (GSM8K, MATH) and out-of-domain code/general reasoning (HumanEval+, MMLU-Pro). Notably, it mitigates performance degradation on non-target capabilities (fewer instances of drops below base model levels, aligning with the goal of preserving pre-trained knowledge.
- The paper's perspective on the SFT-RL gap is relevant and timely. The derivation from policy gradient to token-level loss is well explained and connects to concurrent work like Dynamic Fine-Tuning.

**Weaknesses:**

- OTR bears strong similarities to established regularization methods in LLMs, such as label smoothing, which softens one-hot targets to prevent overconfidence by redistributing probability mass to non-ground-truth classes. More importantly, the penalty term for sampled incorrect tokens (via beta < 0) closely resembles unlikelihood training (Welleck et al, 2019, which is not cited in the paper), where the model is trained to assign lower probabilities to negative samples drawn from its own distribution to combat degeneration; while framed as an RL innovation, OTR could be viewed as a rediscovery or minor extension of these ideas, and the paper lacks direct comparisons to such priors, potentially overstating novelty.
- Despite positioning OTR as bridging the SFT-RL gap through "on-policy" simulation, the single-token rollout and direct penalization of non-ground-truth samples make the "onlineness" aspect vanish in practice. In full RL (e.g., policy gradients for LLMs), on-policy learning involves multi-step trajectories and exploration of novel behaviors; here, rewards are rigidly tied to fixed ground-truth tokens, reducing it to supervised contrastive learning rather than true online adaptation. This limits the method's ability to discover behaviors beyond the initial data distribution, as noted in related work on policy gradients.
- Relatedly, the one-step nature of OTR makes MC rollouts redundant. Given that SFT already provides access to the full model distribution (logits) and target (ground-truth token) at each step, the expectation in Eq. 4 could be computed exactly using the temperature-scaled softmax (\Pi'), e.g., as \Pi'(xt) log \Pi(xt) + \beta ∑_{a ≠ xt} \Pi'(a) log \Pi(a), which is feasible for typical vocab sizes (~100k) on GPUs. This would eliminate sampling variance and cost (K=256), potentially making OTR more efficient and stable. Why use monte-carlo estimation when exact, efficient, closed-form solution is readily available? The paper does not explore or justify why an approximate MC approach is preferred over the closed-form alternative, especially since DFT achieves similar weighting without sampling.
- While DFT is discussed, broader baselines like unlikelihood training or entropy-regularized SFT are absent, weakening claims of superiority.

**Questions:**

- Given the single-token framing, how does OTR handle long-range dependencies in sequences, and does the "on-policy" simulation truly enable exploration akin to multi-step RL, or is it effectively off-policy with model-sampled negatives?
- Why use Monte Carlo sampling instead of exact computation over the distribution? Have the authors experimented with a closed-form version of the loss, and if so, how does it perform in terms of stability and efficiency?

---

> ### Author Response · Authors · 2025-11-27
>
> We thank the **Reviewer z5Y3** for their in-depth review and theoretical insights.
> We appreciate the reviewer's recognition of OTR's empirical performance and our formulation connecting SFT to Policy Gradient. The reviewer raises important theoretical questions regarding the relationship between OTR, Unlikelihood Training, and exact expectation computation, which we address below to clarify our method's distinct position and advantages.
>
> We address the weaknesses and questions below.
>
> ___
> > Weaknesses 1: OTR bears strong similarities to established regularization methods... particularly the penalty term... closely resembles unlikelihood training (Welleck et al, 2019)... OTR could be viewed as a rediscovery or minor extension...
>
> > Weaknesses 4: While DFT is discussed, broader baselines like unlikelihood training or entropy-regularized SFT are absent...
>
> - **Surface Similarity vs. Fundamental Difference:** We acknowledge that OTR shares a structural characteristic with Unlikelihood Training (UT) in that both penalize incorrect tokens. However, this "push-pull" mechanic (increasing probability of positives, decreasing negatives) is a universal feature of Contrastive Learning and Reinforcement Learning (where low-advantage actions are suppressed), not a unique signature of UT. Conflating OTR with UT based on this shared trait overlooks the fundamental differences in **how** negative samples are selected and **why** they are penalized.
>
> - **Divergent Mechanisms (The Core Difference):** The defining difference lies in the source of negative samples. **Unlikelihood Training (UT):** Typically selects negatives from the **previous context** (or high-frequency tokens) specifically to penalize repetition. It is a spatially-grounded heuristic. **OTR (Ours):** Selects negatives via **on-policy Monte Carlo rollouts** from the model's current distribution. This is a probabilistically-grounded approach designed to simulate the RL exploration process. OTR's core contribution is utilizing this **on-policy** signal to improve generalization, which is theoretically distinct from UT's heuristic of suppressing context repetitions.
>
> - **Relevance of Objectives:** UT was proposed in 2019 primarily to solve **neural text degeneration (repetition loops)** in older generation models. In the context of modern LLMs (2025) and reasoning tasks, basic repetition is no longer the primary bottleneck, and UT is rarely used as a standard component in modern SFT pipelines. OTR, conversely, addresses the **generalization gap** between SFT and RL—a critical, unsolved problem in the current era. Applying an outdated repetition-penalty method (UT) as a baseline for a reasoning generalization task would lack practical significance.
>
> - **Justification for Baselines:** We prioritized the comparison with **Dynamic Fine-Tuning (DFT)** because it is a direct concurrent work that, like OTR, specifically investigates the SFT-RL generalization gap. DFT and OTR operate on the same scientific frontier (improving SFT via RL principles), whereas UT and entropy regularization address orthogonal or solved problems (repetition) that do not align with our study's focus on reasoning capabilities.

---

> ### Author Response · Authors · 2025-11-27
>
> ___
> > Weaknesses 2: Despite positioning OTR as bridging the SFT-RL gap... the single-token rollout... make the "onlineness" aspect vanish in practice... reducing it to supervised contrastive learning rather than true online adaptation...
>
> > Question 1: Given the single-token framing, how does OTR handle long-range dependencies... and does the "on-policy" simulation truly enable exploration...?
>
> - **Dynamic Reward Signal vs. Static Supervision:** We respectfully clarify that OTR is not merely SFT with a penalty term. In SFT, the ground truth (GT) is a static target that forces the model distribution regardless of the model's current state. In OTR, the GT serves as a **reward signal** contingent on the model's *current* policy:
>
>   **Hit (GT is sampled):** If the model's current distribution can "reach" the GT, we reinforce this valid path.
>
>   **Miss (GT is not sampled):** If the model fails to sample the GT (meaning the GT is outside the effective exploration region), we penalize *all* sampled tokens.
>
>   This mechanism creates a **conditional update** dynamic: we only reinforce behaviors the model can currently self-generate, while suppressing confident hallucinations. This is fundamentally different from static contrastive learning, as the set of negatives and the magnitude of the gradient are dynamically determined by the model's real-time rollout.
>
> - **Stability and Generalization:** The "on-policy" nature ensures that updates are strictly constrained to regions the model can already reach. As noted in the paper line 56, this prevents the "drastic and potentially harmful shifts" often seen in SFT, where the model is forced to memorize low-probability samples, destroying its pre-trained manifolds (catastrophic forgetting). OTR's exploration is thus a "safe" exploration: it expands the policy's coverage towards the GT only when the policy is locally ready, leading to the superior generalization and reduced forgetting observed in our experiments.
>
> - **Handling Dependencies:** While the *rollout* is single-step, the *optimization* occurs sequentially across the entire trajectory. Improving the policy $\pi_{\theta}(\cdot|s_t)$ at step $t$ via OTR inherently improves the likelihood of generating valid states $s_{t+1}$ for subsequent steps. By optimizing the "local" exploration at every step of the chain, OTR effectively guides the global trajectory without the high variance and credit assignment difficulties of full sentence-level RL.

---

> ### Author Response · Authors · 2025-11-27
>
> ___
> > Weaknesses 3: The one-step nature of OTR makes MC rollouts redundant... the expectation in Eq. 4 could be computed exactly using the temperature-scaled softmax... Why use monte-carlo estimation when exact, efficient, closed-form solution is readily available?
>
> > Question 2: Why use Monte Carlo sampling instead of exact computation over the distribution?
>
> - **Computational Cost Re-verification:** First, we wish to empirically address the concern that Monte Carlo (MC) sampling introduces computational redundancy or inefficiency. We recorded the exact training runtime and speed for both SFT and OTR on the OpenR1 dataset. As shown in the table below, the training time difference is **negligible** (approx. 1.0x cost). **Reason for Identical Speed:** OTR requires **only one single forward pass** per batch (same as SFT) to compute logits. The subsequent sampling of $K$ tokens is a **parallelized tensor operation** executed simultaneously, incurring virtually zero latency compared to the heavy model computation. **Reason for Identical Memory:** The sampling process is wrapped in a `torch.no_grad()` context. This ensures that no computational graph or intermediate gradients are stored for the rollout branch, keeping the peak memory usage consistent with SFT. Thus, the "efficiency gain" of moving to a closed-form solution would be practically non-existent.
>
>
> | Model | Method | Epoch 1 (Total Time \| Speed) | Epoch 2 (Total Time \| Speed) |
> | :--- | :--- | :--- | :--- |
> | **Qwen2.5-7B** | SFT | 1h 01m 01s \| 46.94s/it | 1h 06m 18s \| 51.01s/it |
> | **Qwen2.5-7B** | **OTR** | 1h 01m 29s \| 47.30s/it | 1h 06m 49s \| 51.42s/it |
> | **Qwen3-8B** | SFT | 1h 13m 11s \| 56.30s/it | 1h 27m 21s \| 67.20s/it |
> | **Qwen3-8B** | **OTR** | 1h 12m 38s \| 55.88s/it | 1h 18m 38s \| 60.50s/it |
> - **Defining "On-Policy" Region:** regarding the methodological choice, the core philosophy of OTR is to restrict updates to the "regions the model can already reach." **MC Approach:** Sampling naturally acts as a dynamic filter. If a token is too unlikely to be sampled within $K$ trials, it is effectively treated as "off-policy" and ignored. This aligns perfectly with the RL concept of learning from experienced trajectories. **Closed-Form Approach:** Computing the exact expectation over the *entire* vocabulary (e.g., 100k+ tokens) would include the long tail of irrelevant, near-zero probability tokens. To replicate the "on-policy" behavior in closed-form, we would still need to apply truncation (like Top-K or Top-P) to define the active candidate set.
> - **Hyperparameter Intuition:** Consequently, a closed-form solution is not parameter-free; it essentially trades the sampling count $K$ for a probability threshold parameter. We argue that **$K$ (number of samples)** is a more intuitive hyperparameter in an RL context than an arbitrary probability threshold. While we agree that a truncated closed-form formulation is feasible, we chose MC sampling because it directly simulates the RL exploration process and provides a straightforward control ($K$) over the exploration width, without any computational disadvantage.

---

### Official Review · Reviewer_bCYc · 2025-10-28

**Soundness:** 3
**Presentation:** 3
**Contribution:** 2
**Rating:** 4
**Confidence:** 4

**Summary:**

Proposes a new language model finetuning method, called one-token rollout (OTR), which repurposes supervised finetuning (SFT) data for on-policy RL. The main idea is to use the next token in an output as a reward signal. For each token, instead of applying the standard cross-entropy loss, OTR samples multiple candidate tokens from the current policy and performs a policy gradient update, where the reward of the ground truth token is set to 1 and the reward of other tokens is set to a lower (negative) value. OTR is demonstrated to generalize better than standard SFT across several benchmarks and base models.

**Strengths:**

1. Relatively well-written and easy to follow.

2. The idea behind OTR is conceptually simple. I also find that it is sufficiently motivated through existing observations on the benefits of on-policy data.

3. I appreciate the comparison to the concurrent dynamic finetuning (DFT) method in Section 3.4. It helps clarify the distinction of OTR (despite not being strictly necessary due to concurrent work guidelines of ICLR).

4. The empirical evaluation considers several different benchmarks, including in-distribution and out-of-distribution ones.

**Weaknesses:**

1. It seems that OTR incurs an additional computational cost over SFT due to the need of generating multiple candidate tokens for each token in a response. It would be beneficial to:
    - Report for the existing results what the compute difference (e.g., FLOPS or runtime) was between OTR and SFT.
    - If it is non-negligible, run experiments where both OTR and SFT are allocated the same amount of compute. For example, can let SFT run for more optimization steps until it reaches the same compute that OTR used (based on the same data).
    - Straightforwardly discuss this limitation, specifying the exact computational overhead that OTR incurs compared to SFT.

2. The empirical evaluation compares OTR only to SFT. There are a plethora of methods to finetune language models. Thus, it is difficult to assess the significance of OTR, the sole contribution of this paper. In particular, how does it compare to RL with verifiable rewards (RLVR)? While RLVR employs a different type of supervision (reward on answer vs a demonstration of a reasoning chain and answer), I believe this is still a necessary comparison given that verifiable rewards in the domains considered can be easier to obtain than high quality demonstrations.

3. The gains of OTR over SFT are in many cases quite small. The paper does not report standard deviations, or any other measure of statistical significance. So it is unclear in how many of these cases the difference is just due to noise.



Review Summary and Recommendation
---

Overall, I find this paper to be on the borderline. On the one hand, it is relatively well-written and proposes a simple (in a good sense) method for converting SFT into an on-policy method. However, the empirical evaluation is somewhat lacking. I would be willing to reconsider my score if the authors are able to treat the comments above regarding the empirical evaluation and additional compute required by OTR.


Additional (More Minor) Comments
---

1. In the SFT definition (Equation (1)), conditioning on the prompt is missing. Also, as far as I am aware, the standard formulation does not normalize across the length of the sequence, rather across the number of sequences.

2. The explanation starting from line 151 on the use of temperature for sampling is rather standard. It is a subjective matter, but in my opinion it can be significantly shortened (e.g., there is no need to define the alternative sampling policy and explaining the generation process, rather it is possible to just say that a temperature greater than 1 is used to encourage exploration).

3. I do not believe it is necessary to write a stop gradient around the reward in Equation (8). At least, it is not standard to do so.

**Questions:**

--

---

> ### Author Response · Authors · 2025-11-27
>
> We thank the **Reviewer bCYc** for their thoughtful review and constructive feedback.
> We are glad that the reviewer finds our method conceptually simple, well-motivated, and appreciates our comparison with concurrent work like DFT. We also value the recognition of our extensive evaluation across diverse benchmarks.
>
> We address the weaknesses and questions below.
>
> ___
> > Weaknesses 1: It seems that OTR incurs an additional computational cost over SFT due to the need of generating multiple candidate tokens... It would be beneficial to: Report ... what the compute difference ... was. If it is non-negligible, run experiments where both OTR and SFT are allocated the same amount of compute... Straightforwardly discuss this limitation.
>
> * **Clarification on Computational Complexity:** We respectfully clarify that OTR does **not** incur the high computational cost associated with autoregressive generation. The "sampling" in OTR is performed via a parallelized tensor operation (e.g., `torch.multinomial`) on the logits computed during the **single forward pass** already required for SFT. It does **not** involve generating $K$ independent sequences, which would require multiple forward passes. In our implementation, this sampling is wrapped in a `torch.no_grad()` context, ensuring no additional computational graph or gradient memory is used for the candidates. The memory footprint is dominated by the model weights and the single forward pass, making it **nearly identical to standard SFT**.
> * **Empirical Verification of Equal Compute:** To address the reviewer's request for a compute difference report, we recorded the exact training runtime for SFT and OTR on the OpenR1 dataset. As shown in the table below, the time difference is **negligible** (approx. 1.0x cost). In some runs (e.g., Qwen3-8B), OTR was even faster due to system fluctuations. Since the computational overhead is effectively zero, our current experiments **already satisfy the reviewer's criteria** for "allocating the same amount of compute." SFT is the precise compute-matched baseline for OTR.
>
> | Model          | Method  | Epoch 1 (Total Time \| Speed) | Epoch 2 (Total Time \| Speed) |
> | :------------- | :------ | :---------------------------- | :---------------------------- |
> | **Qwen2.5-7B** | SFT     | 1h 01m 01s \| 46.94s/it       | 1h 06m 18s \| 51.01s/it       |
> | **Qwen2.5-7B** | **OTR** | 1h 01m 29s \| 47.30s/it       | 1h 06m 49s \| 51.42s/it       |
> | **Qwen3-8B**   | SFT     | 1h 13m 11s \| 56.30s/it       | 1h 27m 21s \| 67.20s/it       |
> | **Qwen3-8B**   | **OTR** | 1h 12m 38s \| 55.88s/it       | 1h 18m 38s \| 60.50s/it       |
>
> ___
> > Weaknesses 2: The empirical evaluation compares OTR only to SFT... In particular, how does it compare to RL with verifiable rewards (RLVR)? ... I believe this is still a necessary comparison given that verifiable rewards in the domains considered can be easier to obtain than high quality demonstrations.
>
> * **Computational Fairness:** While we agree that RLVR is a powerful paradigm, comparing it directly to OTR is computationally unfair. As established in our response to Weakness 1, OTR operates with **virtually the same compute budget as SFT** (~1.0x cost). In contrast, RLVR requires autoregressive rollouts of full sequences, which typically incurs training times and memory footprints that are **several times higher** than SFT/OTR. Therefore, SFT remains the most scientifically rigorous baseline to measure the effectiveness of our method under a constrained compute budget.
> * **Methodological Positioning:** We position OTR not as a competitor to full-scale RL, but as a **data-centric evolution of SFT**. Our core contribution is demonstrating that by simulating token-level on-policy interactions, we can "upgrade" the standard SFT process to enjoy generalization benefits typically associated with RL, *without* paying the computational price of RL. Therefore, demonstrating superiority over SFT conclusively proves that OTR utilizes the **same** compute budget more effectively than the industry-standard approach.
> * **Scientific Contribution:** A key contribution of our work is isolating the impact of **on-policy data**. By achieving RL-like generalization improvements using SFT-like compute, OTR provides compelling evidence that the "on-policy" nature of data—rather than complex RL algorithms or reward verifiers—is a critical factor in explaining the generalization gap between SFT and RL.

---

> ### Author Response · Authors · 2025-11-27
>
> ___
> > Weaknesses 3: The gains of OTR over SFT are in many cases quite small. The paper does not report standard deviations, or any other measure of statistical significance. So it is unclear in how many of these cases the difference is just due to noise.
>
> - **Robust Evaluation Methodology:** We explicitly designed our evaluation metrics to mitigate the impact of randomness, as detailed in **Section 4.2**. For smaller, high-variance datasets (e.g., AMC, AIME), we employed a **pass@16 (mean@16)** metric. Averaging over a larger number of samples ($K=16$) drastically reduces the variance and ensures that the reported scores reflect the model's true capability rather than generation noise. For large-scale datasets (e.g., MATH, GSM8K), the sheer volume of test samples naturally provides statistical robustness. A consistent improvement of ~1-2% on a dataset of thousands of problems represents a correction of significant number of instances, which is statistically meaningful.
>
> - **Consistency as Significance:** While the magnitude of gain varies, the **consistency** of OTR's superiority is the strongest evidence against noise. OTR outperforms SFT across: **4 different model architectures** (Qwen2.5-3B/7B, Qwen3-4B/8B), **3 distinct domains** (Math, Code, General Reasoning), **Both In-Domain and Out-of-Domain tasks**. It is statistically highly improbable that random noise would consistently favor OTR across nearly all experimental configurations (see Table 1 and Table 2).
>
> - **Substantial Improvements:** We also respectfully point out that in several key instances, the gains are substantial, not small. For example, on **Qwen3-4B**, OTR improves MATH accuracy by **+11.35%** (63.95 -> 75.30) and GSM8K by **+17.85%** (74.13 -> 91.98) compared to SFT. Even where gains are moderate, OTR successfully prevents the performance degradation (catastrophic forgetting) often observed in SFT, which is a significant contribution in itself.
>
>
> ___
> > Additional (More Minor) Comments:
> > 1. In the SFT definition (Equation (1)), conditioning on the prompt is missing...
> > 2. The explanation starting from line 151 on the use of temperature... can be significantly shortened...
> > 3. I do not believe it is necessary to write a stop gradient around the reward in Equation (8).
>
> - **Acknowledgement:** We sincerely thank the reviewer for their meticulous attention to detail and valuable suggestions regarding the mathematical formulations and presentation. We will carefully consider these points and incorporate necessary revisions in the final version of the manuscript to improve clarity and precision.

---

> > ### Comment · Reviewer_bCYc · 2025-11-27
> >
> > Thank you for the response. I have read it and the other reviews carefully. I appreciate the clarification regarding the computational efficiency of OTR compared to SFT and additional details. The response has largely addressed the concerns raised in my original review.
> >
> > I will note that there is a difference between variability of results due to randomness in sampling during evaluation and variability across randomness in training runs. RL, and potentially OTR due to it being somewhere in between SFT and RL, can often exhibit instability across training random seeds. Thus, I maintain that reporting standard deviation across random seeds for training, at least for some settings, can improve the significance of OTR. Nonetheless, I am increasing my initial rating in light of the authors’ response.

---

### Official Review · Reviewer_hxWy · 2025-10-29

**Soundness:** 3
**Presentation:** 3
**Contribution:** 2
**Rating:** 4
**Confidence:** 4

**Summary:**

This paper describes a method to improve on supervised fine-tuning (SFT) of large language models (LLMs) by doing one-token rollouts at every subsequence of an SFT dataset. The method samples K token completions of at every step of every sequence in the training dataset, and defines a REINFORCE-like loss where the reward is the fraction of K tokens that match the next ground truth token.

Qwen models are fine-tuned using both the one-token rollout (OTR) method and SFT on a math training dataset, and evaluated on a wide suite of math and code generation benchmarks. Results show that OTR generally improves over SFT.

Ablation experiments show the importance of adding a slightly negative reward on tokens that are not equal to the ground-truth next token helps with training stability and improves performance.

**Strengths:**

1. The paper presents a simple way of implementing on-policy REINFORCE-like improvement of LLMs without doing full reinforcement learning, such as using GRPO. The method is quite simple to implement given an SFT training dataset.

2. Results show improvements over SFT fine-tuning for Qwen models, across a wide set of math and code generation benchmarks. The paper reports out-of-domain generalization improvements with OTR training as well.

3. Ablations are included to show the necessity of using a small negative reward for incorrectly generated tokens.

**Weaknesses:**

1. One key missing aspect of this paper is a measure of complexity of the proposed method. In the introduction the paper claims that it's method "improves model generalization without incurring the computational cost of full sentence generation." However, as presented in the paper, for every training set sequence of length T tokens, OTR samples K tokens, which equals a total of T*K sampled tokens. This is the same as sampling K length-T completions for the prompt corresponding to the training set sequence. Note that K=256 is quite high in the paper experiments.

It is difficult to evaluate the claims of this work without any analysis of its additional complexity, which is not in the current paper.

2. Related to Weakness (1), SFT is not the correct baseline for OTR. Since the method takes on-policy samples (and from the above analysis requires TK samples per training dataset sequence), an online RL baseline is more appropriate. For example, an RL run which samples 1 full trajectory per training dataset sequence (requiring T samples), performs an agent update iteration, then repeating this K times, would match the compute used. It would be much more helpful to compare the performance improvement achieved by OTR with this baseline to evaluate the effectiveness of this method.

3. The paper only trained using Qwen models. This limits the evaluation of the generality of the proposed method across model types.

**Questions:**

In "training details" it is mentioned that OTR and SFT were trained using identical settings. Why is this? Shouldn't each setting be optimized in order to obtain the highest improvement?

Why are only Qwen models used? Why not other open-source models with similar number of parameters?

In equation (6), why does the policy \pi have a ' mark?

---

> ### Author Response · Authors · 2025-11-27
>
> We thank the **Reviewer hxWy** for their constructive feedback and detailed review.
> We are encouraged that the reviewer recognizes the simplicity of our method, its effectiveness in improving performance over SFT across math and code benchmarks, and the value of our ablation studies regarding the negative reward signal.
>
> We address the weaknesses and questions below.
>
> ___
> > Weaknesses 1: One key missing aspect of this paper is a measure of complexity of the proposed method... for every training set sequence of length T tokens, OTR samples K tokens, which equals a total of T*K sampled tokens. This is the same as sampling K length-T completions for the prompt corresponding to the training set sequence... It is difficult to evaluate the claims of this work without any analysis of its additional complexity.
>
> - **Clarification on Computational Complexity:** We respectfully clarify that there is a **fundamental misunderstanding** regarding the implementation of OTR. Sampling $K$ tokens at each step in OTR is **NOT** equivalent to generating $K$ full sequences (completions).
> - **No Autoregression:** OTR does **not** involve autoregressive generation (which would require $K$ separate forward passes or a batch size increase of $K$). Instead, like standard SFT, OTR performs **only one single forward pass** per batch to compute the logits for the input sequence.
> - **Parallel Sampling:** The $K$ candidates are sampled directly from these computed logits using a parallelized tensor operation (e.g., `torch.multinomial`) across all positions simultaneously. This is a lightweight mathematical operation, not a model inference cost.
> - **Implementation Efficiency:** In our codebase, the sampling process is wrapped in a `torch.no_grad()` context. We do not construct a computational graph for the sampling steps, nor do we store intermediate activations for $K$ branches. The memory footprint is dominated by the model weights and the single forward pass, making it **nearly identical to standard SFT**.
> - **Empirical Evidence:** To empirically validate this, we recorded the exact training time and speed (seconds per iteration) for both Epoch 1 and Epoch 2 on the OpenR1 dataset mentioned in the paper. As shown in the table below, the time difference between SFT and OTR is negligible. In fact, for the Qwen3-8B model, OTR was slightly faster due to system environment fluctuations, conclusively proving that OTR incurs **virtually zero additional time overhead** compared to SFT.
> | Model | Method | Epoch 1 (Total Time \| Speed) | Epoch 2 (Total Time \| Speed) |
> | :--- | :--- | :--- | :--- |
> | **Qwen2.5-7B** | SFT | 1h 01m 01s \| 46.94s/it | 1h 06m 18s \| 51.01s/it |
> | **Qwen2.5-7B** | **OTR** | 1h 01m 29s \| 47.30s/it | 1h 06m 49s \| 51.42s/it |
> | **Qwen3-8B** | SFT | 1h 13m 11s \| 56.30s/it | 1h 27m 21s \| 67.20s/it |
> | **Qwen3-8B** | **OTR** | 1h 12m 38s \| 55.88s/it | 1h 18m 38s \| 60.50s/it |
>
> ___
> > Weaknesses 2: Related to Weakness (1), SFT is not the correct baseline for OTR. Since the method takes on-policy samples... an online RL baseline is more appropriate... It would be much more helpful to compare the performance improvement achieved by OTR with this baseline to evaluate the effectiveness of this method.
>
> - **Invalidation of Premise:** This concern is premised on the assumption—which we addressed in Weakness 1—that OTR incurs a high computational cost equivalent to generating $K$ full trajectories. As demonstrated by our training logs, **OTR incurs virtually zero additional time or memory overhead compared to SFT** (~1.0x cost). In contrast, Online RL baselines (even simplified ones) require autoregressive generation, which is orders of magnitude slower and consumes significantly more memory for maintaining rollout buffers.
>
> - **Fairness of SFT Baseline:** Because OTR operates within the exact same computational budget and infrastructure as Supervised Fine-Tuning (using a fixed dataset, single forward pass, and no autoregressive rollout), **SFT is the precise compute-matched baseline**. Comparing OTR (a lightweight, single-pass method) to Online RL (a heavyweight, multi-pass/autoregressive method) would be an unfair comparison of methods with vastly different resource requirements.
>
> - **Methodological Positioning:** We position OTR not as a competitor to full-scale RL, but as a **data-centric evolution of SFT**. Our core contribution is demonstrating that by simulating token-level on-policy interactions, we can "upgrade" the standard SFT process to enjoy generalization benefits typically associated with RL, *without* paying the computational price of RL. Therefore, demonstrating superiority over SFT conclusively proves that OTR utilizes the **same** compute budget more effectively than the industry-standard approach.

---

> ### Author Response · Authors · 2025-11-27
>
> ___
> > Weaknesses 3: The paper only trained using Qwen models. This limits the evaluation of the generality of the proposed method across model types.
>
> > Question 2: Why are only Qwen models used? Why not other open-source models with similar number of parameters?
>
> - **Strategic Choice for High-Performance Baselines:** We deliberately focused on the Qwen series (specifically Qwen2.5 and Qwen3) because they currently demonstrate state-of-the-art performance among open-source models in the 3B-8B parameter range. Our primary objective was to evaluate whether OTR provides gains *on top of* the strongest available base models that are most likely to be deployed in real-world applications, rather than demonstrating improvements on weaker architectures where gains might be easier to achieve.
> - **Robustness Across Generations:** While we focused on one model family, we validated OTR across two distinct generations (Qwen2.5 and Qwen3) and multiple sizes (3B, 4B, 7B, 8B). The consistent improvement across these different architectures and scales provides strong evidence that OTR is robust to variations in model depth and architectural nuances.
> - **Future Expansion:** We appreciate the reviewer's suggestion to broaden the evaluation to ensure cross-family generality. We plan to include experiments on other model families (such as Llama-3 or Mistral) in the final version of the paper to further verify the method's universality.
>
> ___
> > Question 1: In "training details" it is mentioned that OTR and SFT were trained using identical settings. Why is this? Shouldn't each setting be optimized in order to obtain the highest improvement?
>
> - **Controlled Comparison:** We maintained identical hyperparameters strictly to ensure a **fair, controlled experiment**. By holding all other variables constant (learning rate, batch size, scheduler), we isolate the performance gain attributable solely to the algorithmic innovation of OTR, eliminating the confounding factor of hyperparameter tuning.
> - **Effectiveness for Both:** The settings reported in the paper are standard, well-established configurations for fine-tuning models. Our preliminary experiments confirmed that these settings allow **both** SFT and OTR to converge smoothly and achieve strong performance. Therefore, the comparison reflects a valid assessment of their relative capabilities under a robust, common operating point.
>
> ___
> > Question 3: In equation (6), why does the policy \pi have a ' mark?
>
> - **Notation Clarification:** The prime symbol ($'$) is used to distinguish the **exploration policy** from the original model policy $\pi_{\theta}$. As explicitly defined in **Equation (5)**, $\pi'_{\theta}$ is derived by applying a temperature scaling (parameter $\kappa$) to the original logits. This transformation flattens the probability distribution to encourage broader exploration of candidate tokens during the rollout phase, ensuring the model evaluates a diverse set of potential next tokens rather than just the most probable ones.

---

### Note · Authors · 2025-12-31

I have read and agree with the venue's withdrawal policy on behalf of myself and my co-authors.